The response of geophytes to continuous human foraging on the Cape south coast, South Africa and its implications for early hunter-gatherer mobility patterns

Botha M. Susan 1 susan.botha@gmail.com
http://orcid.org/0000-0003-3514-2685 Cowling Richard M. 1
De Vynck Jan C. 1
http://orcid.org/0000-0001-6510-727X Esler Karen J. 2
http://orcid.org/0000-0003-0919-7279 Potts Alastair J. 3
1 Botany Department, African Centre for Coastal Palaeoscience , Gqeberha, Eastern Cape , South Africa
2 Stellenbosch University, Conservation Ecology & Entomology , Stellenbosch, Western Cape , South Africa
3 Botany Department, Nelson Mandela University , Gqeberha, Eastern Cape , South Africa
Lipo Carl
Electronic publication date: 2022 May 3
Publication date: 2022
Volume: 10
Electronic Location ID: e13066
Received 2021 Aug 25; Accepted 2022 Feb 14
Copyright: © 2022 Botha et al.
Copyright year: 2022
Copyright holder: Botha et al.
License: This is an open access article distributed under the terms of the Creative Commons Attribution License, which permits unrestricted use, distribution, reproduction and adaptation in any medium and for any purpose provided that it is properly attributed. For attribution, the original author(s), title, publication source (PeerJ) and either DOI or URL of the article must be cited.
License URL: https://creativecommons.org/licenses/by/4.0/

Keywords: Underground storage organs, Harvesting, Bet-hedging, Hunter-gatherer mobility patterns, Geophyte ecology

Funding: South African National Research Foundation (NRF) Doctoral Scholarship National Research Foundation Research Career Advancement Fellowship 91452 NRF of South Africa 93487, CPRR14072880885 Susan Botha was funded by a South African National Research Foundation (NRF) Doctoral Scholarship. Alastair Potts is supported by the National Research Foundation Research Career Advancement Fellowship (Grant No: 91452). This research was also supported by the NRF of South Africa (Grant No: 93487, CPRR14072880885) and Curtis Marean. The funders had no role in study design, data collection and analysis, decision to publish, or preparation of the manuscript.

==============================
Current ecological understanding of plants with underground storage organs (USOs) suggests they have, in general, low rates of recruitment and thus as a resource it should be rapidly exhausted, which likely had implications for hunter-gatherer mobility patterns. We focus on the resilience (defined here as the ability of species to persist after being harvested) of USOs to human foraging. Human foragers harvested all visible USO material from 19 plots spread across six Cape south coast (South Africa) vegetation types for three consecutive years (2015–2017) during the period of peak USO apparency (September–October). We expected the plots to be depleted after the first year of harvesting since the entire storage organ of the USO is removed during foraging, i.e. immediate and substantial declines from the first to the second harvest. However, over 50% of the total weight harvested in 2015 was harvested in 2016 and 2017; only after two consecutive years of harvesting, was there evidence of significantly lower yield (p = 0.034) than the first (2015) harvest. Novel emergence of new species and new individuals in year two and three buffered the decline of harvested USOs. We use our findings to make predictions on hunter-gatherer mobility patterns in this region compared to the Hadza in East Africa and the Alyawara in North Australia.

Introduction

The Cape south coast falls within the Greater Cape Floristic Region (GCFR) (Manning & Goldblatt, 2012; Snijman, 2013), an area supporting 11,423 plant species (80% endemic), making it the richest concentration globally of extratropical plant diversity and endemism (Snijman, 2013). Within the Cape Floristic Region (CFR; a sub-region of the GCFR that excludes the Succulent Karoo component and incorporates the Cape south coast) 1,552 species (17% of total) are geophytes—species with underground storage organs (USOs) (Procheş, Cowling & du Preez, 2005).

Geophytes are defined as plants with an underground perennation organ (bulb, corm, tuber, or rhizome) that act as a storage organ (Procheş et al., 2006). Some geophytic species are evergreen whilst others have leaves that die back annually (a period of annual dormancy). In the fire-driven Cape flora, some USOs not only flower every year, but flowering may be stimulated by fire (Lamont & Downes, 2011). Current evidence suggests most USOs are long-lived and may have poor recruitment (Adler et al., 2014), but this topic is underexplored. These ecological traits of geophytes—annual visibility, emergence post-fire, recruitment rates and resilience to harvesting—would be key factors in determining how reliant hunter-gatherer communities could be on this resource, and would have impacted on hunter-gatherer’s mobility, density and dispersal patterns (Erlandson et al., 2020). For example, the annual “visibility period” (where the leaves/flowers are present above ground), would define the time harvesters are able to see, and thus extract, any given geophyte. In the Cape south coast, it appears that many geophytes have a long visibility period: 8.6 months per annum on average across species (De Vynck, Van Wyk & Cowling, 2016; see Supplemental Material) and show little seasonal differences (apart for Renosterveld) in harvesting yield (Botha et al., 2020), suggesting a long window of harvesting opportunity.

Fire-stimulated flowering—where the geophyte’s underground structure survives fire and then flowers (Lamont & Downes, 2011)—may have been another key ecological trait that early human harvesters exploited. Deacon (1993) proposed that early hunter-gatherers could have “farmed” such geophytes (see Vlok, 2020a for research linking geophyte emergence and increased light) by using fire to “flush out” geophytes. This might have been an exceptionally profitable exercise since many geophytes emerge en-masse after a fire (Le Maitre & Brown, 1992) and “simulated” harvesting shows that carbohydrate returns from harvesting geophytes are up to three times higher in the first few years after a fire (Botha et al., 2020). Historical accounts of the human use of fire to burn natural vegetation on a landscape scale is scant, but early travellers (from about 1400 AD) to the Cape noted Khoe-khoe pastoralists burning vegetation to improve pasture quality for livestock (Skead, 2009).

Geophytes generally do not survive foraging since the entire plant is removed when it is harvested (Botha et al., 2020), and (based on anecdotal observations) a decline in yield of USOs has been observed with continuous harvesting (e.g. O’Connell & Hawkes, 1984; Vincent, 1985). Combined with potentially low rates of recruitment (Adler et al., 2014), it suggests that once an area has been depleted by harvesting, it will remain depleted for a period of time. There must therefore be strong selective pressure on geophytes to safeguard their survival (Lovegrove & Jarvis, 1986), e.g. having adaptations to profit from (dispersal) or avoid being harvested (such as being toxic). Only one study in the GCFR suggests potential dispersal traits for a single geophyte species, Micranthus junceus, by mole-rats; but the study does not outright demonstrate a dispersal adaptation, and the extent to which mammalian predator-USO systems occur (including humans) remain unknown.

However, USO use is a well-documented component of hunter-gatherers’ diets worldwide (Archer, 1982; Hurtado & Hill, 1987; Lee, 2017; O’Connell & Hawkes, 1981; Vincent, 1985) and archaeological evidence of USO use in the GCFR is well established (Archer, 1982; Botha et al., 2019; Liengme, 1987; Parkington & Poggenpoel, 1971; Singels, 2020; van Wijk, 2017; Wells, 1965). Latest research indicates Neanderthals and modern humans’ oral microbes contain bacteria specifically designed to break down starch for at least the last 100,000 years and this starch may have been essential for fuelling the evolution of our larger brains (James et al., 2021).

Recent research into reconstructing the Cape south coast resourcescape suggests that when the intertidal resource—demonstrated to be a key component of diet (Colonese et al., 2011)—is unavailable and game was too sparse or distant from the home base for efficient hunting, USOs likely contributed an important dietary component to prehistoric humans (Botha et al., 2020; De Vynck et al., 2016a; Singels et al., 2016; Wren et al., 2020) to enable humans to persist in this region. However, empirical data on both resource density and resilience of USOs to depletion by human foraging is important to understand how this resource may have influenced early humans’ mobility patterns, economies and how they managed to survive in this region. In an analogous example from intertidal foraging, De Vynck et al. (2016a) showed that the highest-yielding intertidal organism for foragers along the coastal shore of the Cape south coast is the alikreukel Turbo samarticus, which is resilient to repeated experimental harvesting as these gastropods recolonize the intertidal from a subtidal population protected from foragers. There has been no similar systematic research on the response of edible USOs to human foraging on the Cape south coast (or elsewhere), and this study seeks to start filling this gap.

We established 19 marked plots and harvested all edible biomass from the plots for three consecutive years (described in the “Methods” section); foraging was conducted by local people from nearby settlements who still have knowledge about indigenous edible plant species (De Vynck, Van Wyk & Cowling, 2016). To our surprise, edible biomass was only significantly reduced after three consecutive years of harvesting; nonetheless, there remained an unexpectedly high yield in some plots (further explained below). This finding may have implications on how we predict settlement-subsistence patterns of early hunter-gatherers. We discuss possible reasons for our findings below.

Physiographic context

The diversity of taxa with USOs in the Greater Cape Floristic Region (GCFR) is unparalleled globally (~2,100 species, ~17% of total flora) (Procheş, Cowling & du Preez, 2005). Our study area is the same as that used in Botha et al. (2020) to quantify return rates of plant foods from experimental foraging bouts. In brief, it forms part of the coastal forelands of the Cape south coast of South Africa (Fig. 1) and is located between the archaeological sites of Blombos Cave in the west and Pinnacle Point in the east and extends inland to the foothills of the Langeberg-Outeniqua mountains. Blombos and Pinnacle Point caves are of global significance as they have provided evidence of human cognitive complexity (Brown et al., 2009; Brown et al., 2012; Henshilwood et al., 2002; Henshilwood et al., 2011; Marean et al., 2007). The area falls within the Greater Cape Floristic Region (GCFR), an intensively studied biodiversity hotspot (Allsopp et al., 2014). The GCFR contains 11,423 plant species, and 1,119 genera of which 77.9% and 22.2% are endemic, respectively (Snijman, 2013).

Figure 1 The location of 19 permanent plots set up across the Cape south coast to capture a diverse array of USOs in different vegetation types.

The Greater Cape Floristic Region is shown in orange. The lines on the inset figure represent the rainfall regions which are: WRZ, Winter rainfall zone; ARZ, All-year rainfall zone; SRZ, Summer rainfall zone (Chase & Meadows, 2007). Map created by Adriaan Grobler.

Three biomes occur in the study area, namely Fynbos, Renosterveld and Subtropical Thicket (Bergh et al., 2014). Fynbos is an evergreen, fire-prone ‘heathland’ characterized by the presence of restiods (wiry, evergreen graminoids of the Restionaceae and Cyperaceae), fine-leaved ericoid shrubs and shrublets (Ericaceae, Asteraceae, Rutaceae, Thymelaeaceae) and proteoid shrubs (exclusively Proteaceae) (Bergh et al., 2014). The core distribution of the Fynbos is along the spines of the Cape Fold Belt mountains, from the Bokkeveld Plateau at Niewoudtville south to the Cape Peninsula and then eastwards on the inland mountains, and along the coast to Port Elizabeth (Bergh et al., 2014). The occurrence of Fynbos correlates strongly with nutrient-poor (low availability of phosphorous and nitrogen) sandy soils, either acidic sands of quartzite origin or calcareous or leached coastal sands (Cowling, 1984; Cramer et al., 2014). Renosterveld is also a fine-leaved, evergreen, fire-prone shrubland where the heathland-type taxa (Ericaceae, Proteaceae, Restionaceae) are rare or absent, and having a grass-dominated herbaceous layer and high diversity and abundance of geophytes (Cowling, 1990). It occurs on soils that are generally more fertile and finer-grained compared to Fynbos (Bergh et al., 2014; Cowling, 1984).

Fire is a fundamental driving force in both these biomes. Lightning and lesser-so, falling rocks, are natural causes of fires, with most wildfires being unplanned (natural and human) (Van Wilgen et al., 2010). The fire frequency of this region has likely increased in the last few decades due to increased human densities, high invasive plant densities and favourable weather conditions conducive for wildfires (Kraaij & van Wilgen, 2014). Prior to colonialism, and unlike Fynbos that offers low grazing capacity, Renosterveld would have been more heavily grazed because richer soils resulted in more grasses and higher nutritional value of forage (Cowling, 1984). The once extensive populations of animals that utilized this biome have been exterminated and most of the Renosterveld has been ploughed for agriculture, with only 4% of coastal Renosterveld remaining (Kemper et al., 2000). Since few studies were conducted prior to the disappearance of Renosterveld, we are uncertain of the natural fire regimes but it is generally agreed upon that the fire return interval is shorter in Renosterveld (3–5 years) because of faster plant growth and finer fuels compared to Fynbos in the Cape south coast (7–10 years) (Kraaij & van Wilgen, 2014). Our sites were chosen based on several criteria: foremost the presence of edible geophytes that could be harvested, pristine condition of the site (e.g., no, or little presence of human disturbance such as alien plants), and permission from the landowner. We have no records of the fire history at any given site because this has not been documented. For some sites, we could determine the vegetation age since the last fire because of either visible evidence present on site such as charred plant skeletons, protea node counting or from speaking to the current landowner. The sites we include here cover a range of fire histories across the region.

Fire is not a principal ecological driver in Subtropical Thicket (including valley thicket: Vlok et al., 2003) and the vegetation type possesses a large diversity of growth forms, creating a dense canopy of largely evergreen, broadleaved, sclerophyllous, spiny and/or succulent shrubs and low trees up to 3 m in height (Bergh et al., 2014). Strandveld, which grows close to the sea on Late Pleistocene and Holocene aeolianites, is a mosaic of dune fynbos and thicket. Strandveld, which is classified as a component of the Subtropical Thicket biome, is prone to fires but at less frequent intervals than Fynbos and Renosterveld (Strydom et al., 2020).

The study area occurs within the non-seasonal rainfall regime, where rain can fall in any month but with peaks in late winter to spring (August–October) and in autumn (March–April) (Engelbrecht et al., 2014). The mean annual rainfall ranges from 350 to 500 mm (Engelbrecht et al., 2014; Fig. 2). Although rain occurs year-round, evaporation stress is highest in summer resulting in low apparency of USOs when above-ground parts (including flowers) of most species are no longer visible (De Vynck et al., 2016b). Maximum summer temperatures average about 27 °C in January and February, with minimum winter temperatures averaging 5 °C in July (De Vynck et al., 2016b). Total annual rainfall recorded from climate records (Fig. 2) (Riversdale) in the study area was about 29% higher than the 74-year average (433 mm) in the first year of sampling (2015, 560 mm), 12% lower in 2016 (383 mm), and about 28% lower in 2017 (310 mm) (Fig. 2) (See also Supplemental Material, Fig. S1).

Figure 2 Boxplots showing statistics for monthly precipitation of Riversdale spanning from 1878 to 2020.

Annual rainfall for each year of this study is shown separately (2015–2017) and depicted by the different coloured dashed lines.

Methods

We aimed at including a wide range of USO types (see below for different growth forms) that naturally occur across as many vegetation types as possible. Thus, we identified a total of 19 sites, each representing one or more USO species, across six different vegetation types within the study area (Fig. 1). At each site, we established a plot (10 × 10 m), permanently marked at the corners (using metal rods) for repeated sampling. During harvesting, we delineated the boundaries with ropes to constrain the foragers to the designated plot. The plots were foraged during spring (September–October), to coincide with the maximum appearance of edible USOs to foragers (De Vynck et al., 2016b). The necessary permits were obtained prior to harvesting: A plant collection permit was issued by the Western Cape Nature Conservation Board: Permit number 0028-AAA008-00215. Ethical permission for this study was granted by the human research ethics committee of Nelson Mandela University (H15-SCI-BOT-001). Foragers were instructed to harvest all edible USOs in each plot; the final tally, across all plots, included 21 USO species. We also asked foragers to harvest a medicinal plant, Kedrostis nana, from one of the plots. Kedrostis nana is a deciduous vine with a large tuber, and a good surrogate species for Fockea edulis and Dioscorea elephantipes. All three of these species have swollen tubers that can grow quite large, with a slender vine-like growth form aboveground. Fockea edulis and Dioscorea elephantipes have, according to some, become rare in the wild (Coetzee & Miros, 2010) but are still listed as “least concern” on the SANBI redlist (http://redlist.sanbi.org). Both species were widely used by hunter-gatherers in the past (Botha et al., 2019) but could not be found in the study area.

We categorized USOs according to organ type, namely bulb, corm, tuber (soft or fibrous) and root (the last-mentioned comprising a swollen taproot), and general spatial population structure (scattered or clustered) observed in the landscape (i.e., general distribution of these plants in the wider landscape and not restricted to the plots alone). Soft tubers have a soft skin, high moisture and low fibre content; fibrous tubers have a hard, often scaly skin and high fibre content (Singels et al., 2016).

Foragers were recruited informally via researchers who were familiar with inhabitants of some areas, through existing foragers’ acquaintances, and through word of mouth. The subjects resided in the villages of Melkhoutfontein and Bietoe Dorp (Fig. 1). The context of the study was explained to each participant and their written consent obtained. Many residents from these towns have substantial knowledge of the useful indigenous flora of the area (De Vynck, Van Wyk & Cowling, 2016).

During each sampling bout, the foragers harvested edible USOs (this was either known to the foragers or garnered from existing ethnobotanical literature for the region (Botha et al., 2019), taking as much time as needed to retrieve all USO plants from inside the plot. No other edible portions of plants such as seeds, fruits or leaves were collected. Upon foraging completion, we quantified (in the field) the number and edible weight (g) (discarding the sheaths surrounding the corms) of each harvested species. Foragers failed to, or chose not to, harvest all edible USO material on several occasions for various reasons, e.g., very low return rates, inaccessibility, or failing to see a particular plant (Tables S1 and S2). This was determined by the main researcher upon completion of foraging of a selected plot.

We were interested in overall changes of edible biomass from intensive harvesting within a constrained area. Our expectation was that there would be a substantial and statistically significant decline in biomass from the first to the second harvest, or from the first to the third harvest. Thus, we conducted paired parametric and non-parametric tests (Student’s t-test and Wilcoxon test, respectively) to determine if there were significant declines between the first and second, and first and third harvests (we did not conduct any time-series-based statistical analyses as only three time points were available). We used descriptive figures (i.e., box plots and bar-plots) to explore the patterns of changes amongst and within plots.

Results

The total USO weight harvested across all plots and years was 17.2 kg comprising 3,906 individual plants (from a total area of 1,900 m2). The average size range of species removed ranged from 0.06 g to 146.62 g. There was no significant decline in weight harvested between 2015 (411.0 ± 430.3 g [mean ± standard deviation]) and 2016 (372.6 ± 308.6 g) (t = 0.880, df = 18, p = 0.391), but 2017 (187 ± 171 g) had significantly lower weights extracted than in 2015 (t = 2.297, df = 18, p = 0.034) (Fig. 3). In 2017, however, 10 out of the 19 plots had edible weights higher than 50% of 2015s harvest and, of these, five plots had higher returns than the 2015 harvest. USOs with corms and tubers contributed most to the total extracted edible weight (Figs. 4 and 5). Five species (Pelargonium lobatum, Watsonia fourcadii, Chasmanthe aethiopica, Satyrium carneum and Cyphia digitata) comprised 68% of all harvested material while another five species (Albuca fragrans, Babiana tubulosa, Freesia sp., Watsonia sp. and Freesia alba) each contributed less than one percent (Fig. 4).

Figure 3 (A) Total weight harvested from each plot across sampling year. (An outlier point in 2015 is not shown—Plot 10 = 1,907 g). (B) The percentage of the 2015 harvest in subsequent years.

Note that two outliers are not shown on the figure: (A) An outlier point in 2015 is not shown—Plot 10 = 1,907 g, (B) one outlier is not shown in 2016 (Plot 08: 689%).

Figure 4 The total edible weight of underground storage organs harvested each year (during spring) per plot.

The contribution of different species to the total weight is indicated by light grey lines. The shade of the species block shows whether the species was only collected in 1 year (white), across 2 years (grey) or across 3 years (black). The number of species collected in each year per plot is shown in brackets. Plots are ordered according to total yield over the 3 years. Plot 10 represents the Watsonia fourcardii site that had experienced a natural fire in 2014, plot 2 and 19 burnt in 2016. Plot 1 and 14 had experienced a fire approximately 8 and 15 years, respectively, prior to the start of the experiment in 2015.

Figure 5 Proportion of weight harvested (g) across 3 years for each USO plant species across all 19 plots.

Horizontal lines within each year provide an indication of the relative contribution of different plots to the total harvested for each species.

New species and new individuals of species emerged in 2016 and 2017 after the initial extraction in 2015 (Figs. 4 and 5). Corms, roots and bulbs followed a pattern of yearly decline (Fig. 6); however, tubers increased in 2016 and then declined in 2017. This pattern of declining yields was evident in the most abundant species, namely Watsonia fourcadii, Chasmanthe aethiopica (corms dense, clustered populations) and Satyrium longicolle (bulbs in scattered populations) (Fig. 5). The two other high-yielding species also declined but recorded highest harvests in the second rather than the first year of the study, namely Pelargonium lobatum (scattered tubers) and Satyrium carneum (scattered bulbs). No plants of five species (Albuca fragrans, Babiana tubulosa, Oxalis pes caprae, Pelargonium triste and Watsonia sp.) were harvested in the last year of the study. Three species (Babiana ambigua, Kedrostis nana and Pelargonium triste) yielded the highest harvest in the last year, and Freesia alba and Gladiolus teretifolius were only harvested in 2017.

Figure 6 The total weight of USOs harvested per growth form within years across all plots.

Horizontal lines indicate the relative contribution of different USO species summed across all plots.

Discussion

Our expectation was that intense foraging for USOs in a small area during peak apparency would result in a near-complete reduction in biomass extracted for the same period the following year, regardless of the USO growth form and vegetation type. This is because entire plants were extracted during foraging. This expectation was found to be simplistic and naive—there was a wide variation of trends across plots. Below, we aim to describe our findings using the existing, but limited, knowledge on geophytes within the GCFR and elsewhere. This proved to be a difficult task since scientific knowledge on geophyte ecology is in its infancy (within the GCFR and beyond) and only parts of the traditional ecological knowledge (TEK) held by the permanent human residents of this region has survived. TEK can be defined to consist of three parts: (1) there is a component of local observational knowledge of species and other environmental phenomena, (2) a component of practice in the way people carry out their resource use activities, and further, (3) a component of belief regarding how people fit into or relate to ecosystems (Berkes, Colding & Folke, 2000). Much of the first component of TEK: the local knowledge, still exists within the Cape south coast (De Vynck, Van Wyk & Cowling, 2016) and the broader GCFR and parts of it have been documented (Archer, 1982; Botha et al., 2019; Hulley, Van Wyk & Schutte-Vlok, 2018; van Wijk, 2017; Van Wyk & Gericke, 2000; Welcome & Van Wyk, 2019; Youngblood, 2004). However, the way people carry out their resource activities—generally as part of their survival (part two of TEK)—is not widely practiced in the traditional sense anymore, i.e. for survival, or well recorded in the GCFR, and is often “enigmatic, because so much of the content is missing” (van Vuuren, 2016). For example, a song from the/Xam culture (originally Xam speakers occupied large parts of western South Africa) asks when specific plant species will be flowering, for then it will be time to harvest (van Vuuren, 2016). This suggests the/Xam had ecological knowledge about the ideal annual time to harvest a key plant resource. Early hunter-gatherers most likely were experts at such ecological cues—their traditional ecological knowledge “gathered over generations, by observers whose lives depended on this information and its use” (Berkes, Colding & Folke, 2000). Almost all this knowledge has remained unrecorded or has been lost.

The main reason the USO resource was not depleted across species and plots after the initial harvest was the emergence of new species and new individuals in year two and three that buffered the decline of harvestable USOs. We term this staggered emergence: the emergence of a proportion of individuals within USO populations across yearly intervals, i.e., inter-annual emergence variation. The opposite of staggered emergence would be the entire population of any given USO emerging at regular intervals. Inter-annual staggered emergence was the norm for the majority of USO species in the study—eighty percent of geophyte species harvested (19 out of 24 species harvested) varied the timing of their emergence across years, a phenomenon noted elsewhere (Lesica & Crone, 2007; Lesica & Steele, 1994), but not yet for the GCFR. It is noteworthy to mention the study by Boeken (1991) in Israel where emergence of Tulipa systole was not staggered, but insufficient rain meant the new shoots appearing belowground did not have enough resources to emerge aboveground—rainfall received before February increased the probability of the new shoots emerging aboveground.

Staggered emergence buffered the USO weights extracted such that only after three successive annual harvests was there a significant reduction in yield. Yields, however, remained relatively high as a proportion of the first year’s harvest. We highlight a few patterns across individual plots and plant species in general to demonstrate this phenomenon. In plot 18, all apparent individuals of Ferraria crispa were harvested in the first year. The following year, 36% of the total weight extracted from the same plot came from a newly apparent species (Cyphia digitata), which was not present in the first year. In plot 17, all Kedrostis nana individuals were harvested in the first and third year, with no individuals apparent in the second year. For Babiana patula, double the weight (all individuals harvested across all plots) was harvested in year two compared to year one.

Several reasons have been offered to account for such staggered emergence, e.g. climatic vicissitudes and herbivory, and we discuss this below. It has been suggested that inter-annual staggering of above-ground appearance may be a bet-hedging strategy (Slatkin, 1974) to survive either temporal variation in climate (Boeken, 1991; Vidiella & Armesto, 1989) or herbivory (Milchunas & Noy-Meir, 2002), or both (Childs, Metcalf & Rees, 2010). Bet-hedging is theoretically favoured in more variable environments (e.g. differing herbivory intensity or high rainfall variability; Simons, 2011). Simply put, if seed-set fails because of low/variable follow-up rain after flowering, a portion of the USO individuals that did not emerge have the resources to emerge the following year(s) where there is a chance that more favourable conditions might lead to seed-set. Rainfall reliability is most appropriately depicted as the relationship between rainfall amount and variability, quantified as the coefficient of variation (CV) of rainfall amount at a particular time of the year (Bradshaw & Cowling, 2014). Of all the bioregions within the GCFR analysed, the Cape south coast has one of the lowest winter rainfall CV (Bradshaw & Cowling, 2014). This is a trend for the entire GCFR—the winter rainfall CV gradient is exceptionally low across the bioregion’s winter rainfall amount, suggesting even the low rainfall regions such as the Namaqualand semi-desert region have low, but predictable rainfall (Bradshaw & Cowling, 2014; Cowling, Esler & Rundel, 1999). Given the overall low winter rainfall CV gradient across the GCFR in general, a relationship between staggered emergence and rainfall variability might not be very pronounced and should theoretically be low in the Cape south coast. There is a more marked pattern in terms of rainfall seasonality across the GCFR—percentage summer rain increases from west to east within the GCFR and the Cape south coast receives rain year-round. This most likely explains the longer flowering period USOs display in the Cape south coast compared to bioregions located in the strongly winter-dominated rainfall regions of the GCFR (Johnson, 1993).

Bet-hedging by staggering emergence may also be driven by herbivore intensity. The now-submerged Palaeo-Agulhas Plain (PAP), which borders our study area, used to host a novel ecosystem (for the Cape) with a diversity of herbivores, from shrews to elephants (Helm et al., 2018; Klein, 1974; Matthews, Marean & Cleghorn, 2020; Rector & Reed, 2010; Rector & Verrelli, 2010; Thompson, 2010; Venter et al., 2020) during periods of low sea level (Marean, Cowling & Franklin, 2020). This high-biomass, faunal ecosystem was likely underpinned by the high-fertility soils and nutrient-rich grazing (Cowling et al., 2020), and abundance of water (Cawthra et al., 2020) on the inner sector of the PAP. Herbivory intensity would have changed with sea-level fluctuations over time and this variation may have driven the fixation of a bet-hedging genotype within some geophyte species (King & Masel, 2007). Herbivory intensity as an evolutionary driver, is often overlooked in the fynbos region (see Lovegrove & Jarvis, 1986, for a potential adaptation of Micranthus junceus to mole-rat herbivory), although some mammals survived as refuge species and severe extinction rates occurred only during the last 400 years (Venter et al., 2020). Variation in herbivore intensity (Haukka, Dreyer & Esler, 2013) and regional rainfall patterns across the GCFR would be informative in helping to predict human mobility patterns. Regions with longer seasons of USO visibility (where hunter-gatherers can see the plant to harvest it) and where staggered emergence is common, would improve the period of accessibility for hunter-gatherers to utilise this resource. Recent research by Singels (2020) showed that geophyte apparency varies in different regions of the GCFR. In the West Coast, geophyte resource availability is highly seasonal, available only from early winter to early summer whereas in the South Coast and East Coast, geophytes are practically available year-round. This suggests that humans would have favoured the latter two regions during the later summer and autumn months if they required geophytes as a part of their diet.

The resilience of the USO resource would have been an important driver in early hunter-gatherer patterns since humans would have relied on USOs for a source of carbohydrates – deemed an essential part of a hunter-gatherer’s diet (Noli & Avery, 1988; Speth, 1987, 1989; Speth & Spielmann, 1983; Tushingham, Barton & Bettinger, 2021). There is a maximal constraint on the intake of protein by humans, necessitating a diet that incorporates carbohydrates, fat and protein; referred to as “rabbit starvation”—a diet based solely on lean meat can lead to starvation or protein poisoning (Bilsborough & Mann, 2006). Nitrogen metabolized from dietary protein is toxic and must be converted to urea and excreted by the liver, which can only metabolize urea at an approximately fixed rate. An excess production of urea produces physiological stress on the body. This “protein ceiling”—the limit of protein that a human can consume daily is lower for children than adults, for women than men, and for pregnant women (Tushingham, Barton & Bettinger, 2021). The only way to avoid protein poisoning is to substitute foods rich in protein with foods rich in carbohydrates and fat. Although humans have a great capacity for dietary variation, a worldwide survey of ethnographic and modern hunter-gatherer diets by Cordain et al. (2000) showed that hunter-gatherers, despite significant variation in diet, limit their protein uptake to between 19% and 35% of their energetic intake. In the Cape south coast, USOs as a source of carbohydrates, would have been available year-round and unaffected by weather (Botha et al., 2020) and this study was designed to determine how resilient the resource is to harvesting and what influence this might have had on mobility patterns.

It is important to note that natural harvesting behaviour is not constrained to a demarcated plot (Botha et al., 2020; Eder, 1978; O’Connell & Hawkes, 1984) and foraging efforts were intensified in a small space because the task set out was to harvest a fixed area of all edible foods to determine depletion rates. In another study similar to this one, where plant food collection was done without any artificial movement constraints, foragers walked an average of 700 m per hour to harvest food (see Table 2 in Botha et al., 2020). Previously, we also showed that that foraged USO returns are three times higher in recently burnt Fynbos areas (Botha et al., 2020), and thus hunter-gatherers may have used fire to intentionally “farm” USOs, using controlled fire (Bliege Bird et al., 2008). Managing the size of the area burnt and spreading fires over numerous years could have increased the sustainable availability of this resource (Bliege Bird et al., 2008). Whether utilisation was opportunistic in post-fire environments or intentional via “fire-stick farming” (Anderson, 1999; Bliege Bird et al., 2008) remains unknown. The mean annual fire return interval for the region is between 7 and 10 years (Van Wilgen et al., 2010) and could have been at short enough intervals to sufficiently supply hunter-gatherers with enough post-fire environments to exploit without having to employ fire technology. During the 3-year study period, two of our sites experienced a fire. We predict that it would take longer periods to cause depletion around a camp where hunter-gatherers utilize a 10 km foraging radius, than what was experimentally observed in a constrained plot. Furthermore, an updated review on the prevalence of fire-stimulated flowering is needed since the adaptations to fire, including responses to fire intensity and seasonality of fire, must be a more widespread occurrence than the 0.06% of geophyte species within the GCFR documented by Lamont & Downes (2011).

From the perspective of an individual USO species potentially profiting from human foraging, we observed very little recruitment and saw no evidence that geophytes profit from human digging by being dispersed, such as what mole-rats may do when they store USOs in underground caches for later use. Even in a small space with intense foraging, a combination of forager error and discernment appeared to buffer USOs from local extinction. For example, foragers would leave a few individuals behind that they failed to see (forager constraint), or the plants were left behind because they were too small to yield any worthwhile return (plant constraint). The latter behaviour is predicted by optimal foraging theory: humans should forage in a way that maximises their net intake of energy (Hawkes, Hill & O’Connell, 1982; Pyke, 1984; Pyke, Pulliam & Charnov, 1977).

Our study showed depletion, but not complete local extinction, of any USO species in response top repeated human foraging. Anecdotal observations on how geophytes within the GCFR may survive herbivory include the following: USO species that are shallow-rooted (sitting just below the ground making it easy to dig up and extract) survive herbivore extraction of their storage organ by investing in poisonous compounds that make them unpalatable (Harborne, 1991; Mithöfer & Boland, 2012). Edible geophytes are found deeper down in the soil profile, making it harder to forage (Singels, 2020). Another adaptation may be to have multiple sheath layers to protect the edible portion. One species, Ferraria crispa, stacks root corms one on top of the other belowground. It is difficult to harvest all corms, especially in hard soil, and the rest remain behind ensuring survival of the plant. The large-cormed Watsonia fourcadii, which is hyper-apparent in the post-fire environment, is an evergreen geophyte, up to 1-m tall and it grows in clumps in sand fynbos. Significantly higher numbers of plants, as well as flowering plants, appear after clearing of vegetation, through either bush cutting or fire (Vlok, 2020b). In extensive foraging experiments, Watsonia fourcadii (~0.5 kg/h) ranked 7th of all edible species (90 in total) harvested (Botha et al., 2020) and foraging returns of edible geophytes are three times higher in sand fynbos the first 2 years after fire (Botha et al., 2020). In this study the high yield returns for Watsonia fourcadii were corroborated, where a plot to monitor this species was set out in an area that had recently been burned yielded the highest returns of all plots. The archaeological record suggests that humans were aware of this high-yielding genus and utilised it extensively: Watsonia is found in more archaeological sites within the GCFR than any other genus (Botha et al., 2019), and most likely, harvested in post-fire environments where it is more visible, accessible, and productive (Vlok, 2020b).

The Alyawara hunter-gatherers who reside in North Australia, and the Hadza from East Africa, mostly utilize a handful of woody plants as staples, with USOs seemingly less important (O’Connell & Hawkes, 1981; Vincent, 1985). These plants do not have a dormant season, although they are more frequently harvested after rain (see Table 3 for a summary of plants harvested for different hunter-gatherer communities in Botha et al. (2019)). It seems that such woody vines and trees could be sensitive to depletion on the local scale without a seasonal or yearly staggering habit. In O’Connell, Latz & Barnett (1983) study, Ipomoea costata is not found within easy walking distance from residential settlements and the Alyawara would often drive to productive sites and camp there to harvest this species. Prior to local settlements, the Alyawara were highly nomadic and had large foraging ranges (O’Connell, Latz & Barnett, 1983), suggestive that the carbohydrate resource could be sensitive to depletion (Marlowe, 2005). In contrast, the roots and tubers gathered by the Hadza are very productive (Vincent, 1985) with little seasonal variation in returns (Marlowe, 2005) and current-day strain is experienced as a result of their homeland ranges shrinking (Mabulla, 2012). The GCFR presents a unique foraging landscape compared to the Hadza and the Alyawara, where geophytes, the main carbohydrate source, have an exceptionally high diversity, and their life strategies include seasonal, year-long, and fire-stimulated emergence patterns.

Conclusion

In conclusion, we demonstrate that quantifying the depletion of USOs is not as straightforward as predicted. We found significant, but not the expected substantial, declines in harvested USOs and depletion was buffered by staggered emergence of USOs across years. We suggest that humans foragers could have exploited the region for many years before depleting any area substantially. Hunter-gatherers would have benefitted from a landscape where the occurrence of fire created a mosaic-pattern of patches to exploit post-fire. Intentional burning might have allowed hunter-gatherers to manage USO resources even further and it would be interesting to explore whether fire was intentionally utilized.

There is a need for further ecological research in general on the unusually diverse geophytes occurring within the GCFR (Procheş et al., 2006) including longevity, adaptations to herbivory (of both above- and below-ground parts) and fire, as well as reproductive strategies.

The cues, whether environmental or physiological, that trigger a proportion of USO individuals to emerge in any given year is not understood and thus emergence patterns are not predictable. We do not know how long it take for a population of any given USO to recover once it has been depleted, but we predict that recruitment via seed would be low for many species, and therefore USO species that rely predominantly on seeds to establish new individuals, might take on the order of decades to re-populate.

Supplemental Information

Supplemental Information 1 Raw data.

Click here for additional data file.

Supplemental Information 2 Code.

Click here for additional data file.

Supplemental Information 3 Climatic diagrams (Walter-Leith) for three stations for the study years (2015–2017) and mean for the station.

Mean temperature and rainfall indicated in right-hand corner of each plot.

Click here for additional data file.

Supplemental Information 4 The reasons foragers left certain plant species/individuals behind in any given plot after a specific foraging event are captured below.

The various reasons were assigned to two categories: Plant or forager constraint.

Click here for additional data file.

Supplemental Information 5 Additional information about each USO species harvested, including the proportion of plants that were not harvest every year (2015–2017).

Click here for additional data file.

To the resident people of the southern Cape, thank you for sharing your natural heritage with me. To Adriaan Grobler for help with the map.

Additional Information and Declarations

Competing Interests

Author Contributions

Human Ethics

Field Study Permissions

Data Availability

Alastair J. Potts is an Academic Editor for PeerJ.

M. Susan Botha conceived and designed the experiments, performed the experiments, analyzed the data, prepared figures and/or tables, authored or reviewed drafts of the paper, and approved the final draft.

Richard M. Cowling conceived and designed the experiments, authored or reviewed drafts of the paper, and approved the final draft.

Jan C. De Vynck conceived and designed the experiments, authored or reviewed drafts of the paper, and approved the final draft.

Karen J. Esler conceived and designed the experiments, authored or reviewed drafts of the paper, and approved the final draft.

Alastair J. Potts conceived and designed the experiments, analyzed the data, prepared figures and/or tables, and approved the final draft.

The following information was supplied relating to ethical approvals (i.e., approving body and any reference numbers):

Ethical permission for this study was granted by the human research ethics committee of Nelson Mandela University (H15-SCI-BOT-001).

The following information was supplied relating to field study approvals (i.e., approving body and any reference numbers):

A plant collection permit was issued by the Western Cape Nature Conservation Board: Permit number 0028-AAA008-00215.

The following information was supplied regarding data availability:

The raw data and R-code file are available in the Supplemental Files.

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
