# Peer review of "The response of geophytes to continuous human foraging on the Cape south coast, South Africa and its implications for early hunter-gatherer mobility patterns"

_PeerJ, doi:10.7717/peerj.13066_

## Round 0.1 · original submission · Minor Revisions

While we were only able to receive 2 reviews for your article, they both reach the same conclusion that the article is well-written, well-argued, and makes a valuable contribution. Both reviewers suggest only minor revisions and provide some detailed advice as to what needs to be addressed for it to be improved. Please go through the reviewer comments and attend to their suggestions -- they all seem reasonable and readily achievable.

Reviewer 1 ·

Basic reporting

This is an excellent paper that makes important contributions to our understanding of the exploitation potential of USOs (underground storage organs) in sub-Saharan Africa. With some revision as suggested below, all relatively minor, I strongly recommend that the paper be published. Most of the suggestions which follow are requests for some additional information; a few are more substantive; all should be relatively easy to complete. I present most of them in the order in which I encountered an issue or question, not in terms of any order of priority. A few of my comments early on are more general issues that came to mind after reading the entire manuscript. To help the authors find the precise place in the manuscript to which I am referring, I extract a direct quote from their text, flagged in quotation marks, followed by my comment in brackets.

[O'Connell is misspelled throughout the paper.]

"The Cape south coast falls within the Greater Cape Floristic Region (GCFR) (Manning & Goldblatt 2012; Snijman 2013), an area supporting 11,423 plant species (80% endemic), making t the richest concentration globally of extratropical plant diversity and endemism (Snijman 2013). Within the Cape Floristic Region (CFR; a sub-region of the GCFR that excludes the Succulent Karoo component and incorporates the Cape south coast) 1,552 species (17% of total) are geophytes—species with underground storage organs (USOs) (Procheş et al. 2005)."

[Given the unusual nature of the area’s flora compared to other areas of Africa, it might be worth commenting briefly on why this study might still provide relevant comparative insights for other areas where early hominins evolved and used USOs, such as East Africa.]

[It would be helpful to the reader to add a brief discussion of two additional topics: (1) the presumed nature of the natural fire regime (e.g., frequency of lightning-ignited fires in comparison to other arid or semi-arid regions of Africa such as in East Africa. In other words, is the natural fire regime in any way typical or unusual? (2) Perhaps more important to the overall argument is the history of the intensity of grazing by domestic livestock especially since European conquest and colonization. In short, what is the nature and degree of grazing or over-grazing by domestic livestock and how, if at all, might that impact the nature and results of your study? (3) You also need to say something about government policies concerning fire suppression. In the USA the government systematically suppressed natural fires, and others, for decades, with catastrophic consequences. If that has been the case in South Africa to any significant extent, such a policy may conceivably have altered geophyte densities and distributions.]

"We established 19 permanent plots and harvested all edible biomass from the plots for three consecutive years; foraging was conducted by local people from nearby settlements who still have knowledge about indigenous edible plant species (De Vynck et al. 2016c)."

[This quote begins at line 97 in the manuscript. As I read it, I was left with many unanswered questions such as: What do you mean by “established”? Permanent? How big are they? Did you plant geophytes or did you simply fence off an existing natural stand? What sort of enclosure? Why 19? What do you mean by “edible biomass.” Was that edible in your judgment or in the judgment of the villagers who did the harvesting? Just USOs, or seeds, leaves, stems, and other edible biomass as well? How did you determine that you harvested “all”? Did the enclosures keep out grazing ungulates or other animals that might target above-ground or below-ground parts of geophytes? Reading through the rest of the manuscript, not surprisingly, you do address all or at least most, of these questions. So what is needed in the section that begins on line 97 is a statement telling the reader to be patient as the details in answer to these unexplained statements will be provided below....]

"To our surprise, edible biomass was only significantly reduced after three consecutive years of harvesting—yet nonetheless there remained an unexpectedly high yield in some plots."

[What constitutes “significant”? Again, if this is addressed later in the paper, tell the reader that further explanation is coming below.]

"Blombos and Pinnacle Point caves are of global significance as they have provided some of the earliest evidence of human cognitive complexity (Brown et al. 2009; Brown et al. 2012; Henshilwood et al. 2002; Henshilwood et al. 2011; Marean et al. 2007)."

[I know statements like this are popular in some circles, but I am bothered by the phrase “earliest evidence of human cognitive complexity.” Cognitive complexity is obviously a relative concept and is only meaningful in comparison to something else. Hence, one could easily argue that Australopithecines show the earliest evidence of cognitive complexity if your comparison is with fleas. And I suspect fleas show greater cognitive complexity than earthworms, and so forth. In any case, I would rephrase this. I would have only minor qualms if one states that they had "increased" cognitive capacity, but drop the word "earliest." The earliest came when the living organism for the first time was smarter than the average rock and that was probably some time very early in the Precambrian!]

"The study area occurs within the non-seasonal rainfall regime, where rain can fall in any month but with peaks in late winter to spring (Aug-Oct) and in autumn (Mar-Apr)(Engelbrecht et al. 2014)."

[The lack of seasonality makes this region different from East Africa where most studies that talk about USOs have been conducted (e.g., among the Hadza of Tanzania). You need to say something about why and how this difference does or does not matter when drawing general implications for early human use of geophytes.]

"Kedrostis nana is a deciduous vine with a large tuber, and a good surrogate species for Fockea edulis and Dioscorea elephantipes, two well-known edible USOs that have a similar growth form."

[Why is it a good surrogate? Explain.]

"The main reason the USO resource was not depleted across species and plots after the initial harvest was the emergence of new species and new individuals in year two and three that buffered the decline of harvestable USOs."

[Your explanation of this seems to conflate two different issues. Staggered emergence would seem to be most relevant to members of the same species. Reduced competition might be a factor involved in the emergence of new or different species, although staggered emergence could also be involved.... As I continued reading that entire section, I kept saying to myself "but what about....?" The flow of the arguments would be clarified if right at the beginning of this entire discussion you had something like: There are x number of reasons that have been offered to account for such staggered emergence: (1) climatic vicissitudes; (2) herbivory; (3) etc.... That would tell the reader at the outset where you are going with the discussion that follows.]

"Resource resilience is an important driver in determining early hunter-gatherer mobility patterns. De Vynck et al. (2020) work on determining the resilience of the intertidal shellfish resource of this region, showed that the resource, largely comprising Turbo samarticus (92%), is exceptionally resilient, nutritious and productive (De Vynck et al. 2016a)."

[Here I will reflect my own biases and leave it up to the authors' judgement as to whether and how they might want to handle (rephrase?) this section. Personally I don't think food, either what one eats or how one gets it, has anything whatsoever to do with the emergence of "cognitive complexity." But taking this a step further, I instantly reacted with skepticism to the use of the word "nutritious" when applied to shellfish. It's a very slippery term in general and I got curious about what was supposedly so nutritious about Turbo. So I took a bit of a "deep dive" into the literature to see what that might mean. The results are interesting. As shown by Mclachlan, Anton, and H. W. Lombard, 1980. Seasonal Variations in Energy and Biochemical Components of an Edible Gastropod, Turbo sarmaticus (Turbinidae). Aquaculture 19(2):117-125, the dry weight percentage of protein in these gastropods is over 70%. That's a whopping amount (which nutritionally is a negative, not a positive, Kim Hill notwithstanding). If, as has been argued in numerous studies and venues, humans are limited to a daily intake of protein of less than about 35% of total calories (actually an absolute daily limit in grams of protein per kg body weight), that means that the value of the gastropod as a food is limited no matter how abundant, available, and resilient they are, or how many calories they produce when burned in a food analytics lab. In other words, resilience matters, but only up to a point. This means that well over half of the daily diet of these coastal foragers (i.e., more than roughly 65% of their calories) cannot come from those lovely little gastropods or they would likely suffer from what explorers in the arctic called "rabbit starvation." The high protein content of the gastropods actually increases the likely importance of starchy and oily plant foods and hence the likely value of your geophytes. Interestingly, one of the earliest statements about this issue came from South Africa and was written in part concerning shellfish use: Noli, Hans Dieter, and Graham Avery, 1988. Protein Poisoning and Coastal Subsistence. Journal of Archaeological Science 15(4):395-401. Those authors in their abstract made the following comment: "Current hypotheses concerning coastal palaeodiets are based on excessive protein intake and do not deal adequately with this problem." That was 1988. The situation seems largely unchanged today! Nutrition and calories obviously aren’t the same thing. You need all three macronutrients, protein, fat, and carbs, plus a handful of micronutrients (vitamins, minerals, etc.). Marine mollusks are generally high in protein (Turbo seems typical in that regard), and most are pretty poor in fat and low to modest in carbs. So you may get a fair number of calories from them but might still die if they form the mainstay of your diet. The geophytes therefore could have been absolutely critical to coastal foragers even if their total calorie yield was not as high as you might like or expect. You simply can’t live on just protein. The enzymes in your liver can’t cope with all the nitrogen that is produced by deaminizing the protein. You have to have a diet with some 70-75% of calories from sources OTHER THAN protein. And that’s a lot! I’m sorry but both for these nutritional reasons and just on the logic of the arguments, I simply don’t believe that gastropods had anything to do with making humans cognitively modern. Excessive dependence on them would probably have made humans under-nourished or maybe even starving or dead, but not cognitively superior. They make a great supplement to an overall healthy diet, but they are just that, a supplement, they don’t make a whole diet, not even close. If you could somehow boil them, skim off the fat and carbs, and throw away the bulk of the protein, you’d have a more valuable food, but that would require a huge labor input, untold masses of gastropods, and it probably isn’t technically feasible in the first place. And if you are one who thinks that big game hunting is driven more by costly signaling and prosocial motives, as do Hawkes, O'Connell, and Bliege Bird, then little marine gastropods won't get you very far as a "public good." I think Marean and colleagues are barking up the wrong tree if they try to link shellfish to modernity. Early modernity may have emerged in Southern Africa but I seriously doubt that shellfish, any more than baobab fruits or seeds in Tanzania, or catching bunnies in Europe, had any causal role in it..... So what am I asking the authors to do????..... I think the geophytes offer an important, perhaps critical, source of carbs to augment a diet which may have faced problems posed by resources such as shellfish which contained very high levels of protein and limited amounts of non-protein calories. In my opinion, that critical supplemental roles can and should be emphasized. As written, the paper bemoans the fact that the geophytes aren't as resilient as the gastropods. In my opinion they are as valuable if not more so than those snails. A life based overly heavily on snails would have led to quick extinction, not cognitive complexity..... For discussion of upper protein limits, see for example Cordain et al., 2000. Plant-Animal Subsistence Ratios and Macronutrient Energy Estimations in Worldwide Hunter-Gatherer Diets. American Journal of Clinical Nutrition 71(3):682-692; or Speth, JD, 2010. The Paleoanthropology and Archaeology of Big-Game Hunting: Protein, Fat or Politics? Springer, New York.]

"Secondly, once the resource has been depleted (over a period of two to three years), it would take much longer than that of Turbo samarticus to recover a plant species’ population to its former densities. We observed very little/no recruitment and saw no evidence that geophytes profit from human digging by being dispersed, such as what mole-rats may do when they store USOs in underground caches for later use. However, indirectly, persistence of geophyte species after human foraging was noted. Natural harvesting behaviour is not constrained to a demarcated plot (Botha et al. 2020; Eder 1978; O'Connel & Hawkes 1984) and foraging efforts were intensified in a small space because the task set out was to harvest a fixed area of all edible foods to determine depletion rates. Even in a small space with intense foraging, a combination of forager error and discernment appeared to buffer complete USO extinction. For example, foragers would leave a few individuals behind that they failed to see (forager constraint), or the plants were left behind because they were too small to yield any worthwhile return (plant constraint)."

[I think the above gets the authors in unnecessary difficulties. No foragers in their right mind operated in 10 x 10 m squares. The authors are well aware of that and yet this statement seems to ignore that fact and emphasizes the fact that the 10 m squares got depleted, and the geophytes didn't disperse, and life looks bad for the geophyte forager. I think it should be pretty obvious that the whole region wasn't depleted in three years, just those 10 m2 plots. Later you cite Bliege Bird's study of fire-stick farming in Australia. Why not weave those sorts of models into this discussion right from the get-go? What your study shows is that the 10 m plots get depleted so to use the resource effectively over the long-term one has to use the overall landscape and presumably burn it in a staggered and mosaic pattern much like what Bliege Bird observed in Australia. Under that sort of exploitation regime, it might provide a quite stable and predictable resource and hence an important source of carbs to complement the protein sources typical of shellfish from the coast.]

"Alyawara gatherers in Australia, whose traditional ways of foraging for food are still partially intact, corroborate this prediction—they ignore low-ranked foods where returns fall below energetic efficiency (O'Connel & Hawkes 1981; O'Connel & Hawkes 1984)."

[True, but you are citing rather outdated studies. These same authors have gone on to write many articles on costly signaling and prosocial behavior, arguing that men hunt for big game largely for political or prosocial reasons (prestige, displays of reliability and dependability, generosity, skill, knowledge, fostering cooperation), often deliberately bypassing smaller game. As a result they often come home empty-handy when in fact they could have come home with food. For a fairly recent example, see Hawkes, Kristen, James F. O’Connell, and Nicholas G. Blurton Jones, 2014. More Lessons from the Hadza About Men’s Work. Human Nature 25(4):596-619. Bliege Bird has also written lots on the same topic. See for example Bliege Bird, Rebecca L., and Eleanor A. Power, 2015. Prosocial Signaling and Cooperation Among Martu Hunters. Evolution and Human Behavior 36(5):389-397. In other words, these authors have gone on to emphasize that male hunting is first and foremost about social and political goals, not calories or family provisioning. The hunters commonly bypass small game, not because it has fewer calories, but because it has less value as a “public good” when shared with other males, often other unrelated males.]

"We found significant, but not necessarily the expected substantial, declines in harvested resources, suggesting humans could re-visit the same location a few times. The cues, whether environmental or physiological, that trigger a proportion of USO individuals to emerge in any given year is not understood and thus emergence patterns are not predictable."

[This conclusion as phrased is once again misleading and undermines the importance of this study. I don’t think any forager in their right mind would return year after year to the same 10 x 10 m plot. The authors are well aware of this. So why write it this way? Instead, thought of as a region, esp. if the foragers burned it in some sort of mosaic fashion à la Bliege Bird in Australia, the geophyte resource could likely have been repeatedly harvested year after year. In fact, your study builds a great case that they would have to use it in that staggered mosaic fashion. One archaeological goal will be to figure out when landscape burning that is not natural but cultural gets underway in the region (e.g., via fly ash in pollen cores). That would be an interesting key. Perhaps such data already exist? Another interesting question is how long it would take for an individual 10 m2 depleted plot to recover? Are we talking about a few years? Decades? Longer? Can you at least speculate on that?]

Experimental design

Excellent

Validity of the findings

Excellent

Additional comments

Nothing additional. See basic comments above.

·

Basic reporting

Source documents and information: Written consent was obtained and this is noted in manuscript, but there is no copy of form uploaded; if PeerJ required this, then the authors may attend to it. Similarly, the Metadata sheet on “DepletionForaging” workbook seems to not be fully filled out.

Experimental design

No comment.

Validity of the findings

No comment.

Additional comments

This is a clearly-written presentation of a very useful study. There are not enough studies with this level of detail to actually test hypotheses that abound in the archaeological literature about foraging behavior. The authors make a strong case about the lack of quantitative data available for understanding the impacts of human foragers on plant yields, especially in the GCFR. The study integrates nicely into existing (and especially recent) literature, and it was absorbing and informative to read. It is an impressive study with an enormous dataset, and the findings are robust and offer useful data for future work. I find almost nothing to critique in the way the study was designed, the way it is contextualized in the literature, the way the analyses were conducted, or the way it was presented. The introductory material and discussion was engrossing and informative, especially considering how plant remains are under-researched in hunter-gatherer archaeology. The idea that the evolution of bet-hedging strategies during periods of exposed Palaeo-Agulhas Plain and increased herbivory, leading to fixation in the modern day when it is less necessary, is fascinating. If this is true, then the authors may be picking up on one of the key niche-constructive behaviors of our early ancestors.

I have only a few minor comments, below, and then I believe it is ready to be accepted.

1) Implied linkage of genetic ancestry and traditional knowledge: On Line 176 the authors write, “Many residents from these towns have genetic ancestry linking them to the Khoe-San (de Jongh 2012). The Khoe-San share descendants with the San, who were traditionally hunter-gatherers, and the Khoe-khoen, who were traditionally pastoralists (Crawhall 2006; Schlebusch 2010). Recent research suggests that these people were the direct descendants of Homo sapiens sapiens (Pickrell et al. 2012; Schlebusch et al. 2020; Veeramah et al. 2012) who lived on the Cape south coast since about 160 000 BP (Brown et al. 2009; Marean 2010).” My first critique is that the genetic ancestry of a population is irrelevant to the problem at hand, which is how quickly foragers will deplete a plot of USOs. The way the sentence is set up, it sounds like the authors are implying that because there is some genetic ancestry linking the individuals in their study to ancient foragers, these specific people should have particularly pertinent knowledge about the local environment and foraging opportunities in the present day. This may or may not be true that they do, but the genetic connection does not give any special reason on its own that it should be true. When the authors go back in time to point out that genetic studies broadly – and I do mean very broadly – link living Khoe-San and Khoe-khoen to ancient individuals associated with foraging material culture, and that such populations have likely been in place across southern Africa for more than 100,000 years, they are stretching this argument to its very limit. To me, the entire set of sentences should be removed because it is irrelevant to the point, and risks implying some kind of genetic-cultural relationship between genes and foraging knowledge. Furthermore, have these specific people in these specific towns actually had their genes linked to ancestral or living San? Or is this just an assumption based on what is more broadly known about the distribution of genetic ancestry across southern Africa and (undoubtedly) some assignment of ancestry based on phenotype? It is enough to just say the following: There has been evidence of early human foragers (why even go into the species and subspecies?) in the area going back at least 160,000 years (CITE). And if they want to talk about genetics, they could say something like: Genetic data indicate some continuity between living people across the southern African coast with ancient foragers, thus suggesting that cultural knowledge about USO harvesting in these ecosystems may also have had the potential to be passed down from very ancient times. In short, this section doesn’t add anything to the argument about foraging and risks opening up problems.

2) There are a few random places where some typos remain, but they are far and few between (e.g. no need for comma on line 321 after “region”, line 368 should read “an evergreen geophyte”, etc.). Overall, it is written very clearly and concisely and is very information rich, and would benefit from one last read-through.

3) Line 42 – Is poor reproductive success the best wording? It seems more that they authors are describing the slow growth and difficulty in recruitment, which would only translate to poor reproductive success when there is high herbivory or other reasons for high extrinsic mortality. Perhaps the authors mean fertility?

4) Line 300: suggest adding the following reference, as that is where : Rector, Amy L., and Kaye E. Reed. "Middle and late Pleistocene faunas of Pinnacle Point and their paleoecological implications." Journal of Human Evolution 59.3-4 (2010): 340-357.

5) Lines 358-359 – spelling is “O’Connell

6) Line 410 – reference seems incorrect

7) Figure 1 – It would be useful if this figure showed the distribution of the various vegetation types within the GCFR on the main map. Also, on the tiny map inset it is difficult to discern the GCFR at all. I suggest zooming in on southern Africa – it’s still big enough that a reader can identify where it is. That way the entire GCFR can be shown, and possibly even some lines drawn to indicate the summer vs. winter rainfall in the GCFR for reference for readers less familiar with this specific rainfall configuration.

8) Figure 4 –The caption reads that we are looking at yields of species, but then at first I could see only total weights and years and not discern where the species were. After much effort (which is not really what you want in a figure, which is meant to provide clarity), I think I can see how to interpret it. However, are there some mistakes? For example, how can there be any black boxes in some years, like in Plot 20, but not in all years, if black represents a taxon that was harvested for all three years?

9) Figure 5 – It is almost impossible to discern the difference in the two greys that are used. This is, however, an incredibly useful figure.

10) I would find a single master figure to be very useful that shows the rainfall, year, and yield of each plot together in one place. Even just mapping rainfall onto Fig. 3 would be useful so we can see the relationship.

---

## Round 0.2 · accepted · Accept

Thank you for making revisions and attending to the comments made by the reviewers.